# Fire weather index: the skill provided by ECMWF ensemble prediction system

Francesca Di Giuseppe[1], Claudia Vitolo[1], Blazej Krzeminski[1], Christopher Barnard[1], Pedro Maciel[1], and Jesús San-Miguel[2]

[1]ECMWF, Reading, UK
[2]ICTP, Ispra,Italy

**Correspondence:** F.DiGiuseppe@ecmwf.int

**Abstract.** In the framework of the EU Copernicus program, the European Centre for Medium-range Weather Forecast (ECMWF) on behalf of the Joint Research Centre (JRC) is forecasting daily fire weather indices using its medium range ensemble prediction system. The use of weather forecast in place of local observations can extend early warnings up to 1-2 weeks allowing for greater proactive coordination of resource-sharing and mobilization within and across countries. Using one year of pre-operational service in 2017 and the fire weather index (FWI) here we assess the capability of the system globally and analyze in detail three major events in Chile, Portugal and California. The analysis shows that the skill provided by the ensemble forecast system extends to more than 10 days when compared to the use of mean climate making a case of extending the forecast range to the sub-seasonal to seasonal time scale. However accurate FWI prediction does not translate into accuracy in the forecast of fire activity globally. Indeed when all 2017 detected fires are considered, including agricultural and human induced burning, high FWI values only occurs in 50% of the cases and are limited to the Boreal regions. Nevertheless for very large events which were driven by weather conditions, FWI forecast provides advance warning that could be instrumental in setting up management and containment strategies.

## 1 Introduction

The prediction of fire danger conditions allows fire management agencies to implement fire prevention, detection, and pre-suppression action plans before fire damages occur. However, in many countries fire danger rating relies on observed weather data which only allows for daily environmental monitoring of fire conditions (Taylor and Alexander, 2006). Even when this estimation is enhanced with the combined use of satellite data, such as hot spots for early fire detection, and land cover and fuel conditions it normally only provides 4- to 6-hour warnings. By using forecast conditions from advanced numerical weather models, early warning could be extended up to 1-2 weeks allowing for greater coordination of resource-sharing and mobilization within and across countries. Due to the improved skills of weather forecasting, the use of numerical weather prediction offers a real opportunity to enhance early warning capabilities (Roads et al., 2005; Mölders, 2008, 2010). In recent

years institutions such as Natural Resources Canada (NRC) and the US National Oceanic and Atmospheric Administration (NOAA) have implemented regional fire danger forecasting systems based on their operational weather forecasts (Bedia et al.,

2018). The Global Fire Early Warning System is also an international initiative, promoted by the Canadian Parternship for Wildland Fire Science and the United Nation Office for Disaster Risk Reduction, to provide fire danger forecast up to 10 days ahead using the Canadian operational weather forecasting system (http://canadawildfire.ualberta.ca/gfews). Parallel initiatives are promoted by the European Commission under the umbrella of the Copernicus Emergency Management Service (CEMS), namely the European Fire Forecast Information System (EFFIS, http://effis.jrc.ec.europa.eu/) and its global counterpart the

Global Wildfire Information System (GWIS, http://gwis.jrc.ec.europa.eu/). Both systems principally rely on the Canadian Fire Weather Index (FWI) (Van Wagner et al., 1974, 1985) to rate fire danger and on numerical weather predictions to provide forecast fire danger information at the European and global levels (San-Miguel-Ayanz et al., 2002).

Systems such as the FWI detect dangerous weather conditions conducive of uncontrollable fires rather than modelling the probability of ignition and fire behaviours. The FWI (developed in Canada) is specifically calibrated to describe the fire be-

haviour in a jack pine stand (*Pinus banksiana*) typical of the Canadian forests. However, its simplicity of implementation has made it a popular choice in many countries and it has shown to perform reasonably well in ecosystems very dissimilar to the boreal forest (Di Giuseppe et al., 2016a; de Groot et al., 2007). The FWI calculation only relies on weather forcings and no information on the actual vegetation status is taken into account. When weather forecasts are used in place of observations, uncertainties can be introduced. Sources of uncertainty can be: (i) the limited knowledge of the initial state and (ii) the mis-

representation of physical processes. In the former case, errors are randomly distributed around the true state (Orrell et al., 2001); in the latter, errors produce systematic deviations from the true state. In both cases, errors in the weather forecast may be amplified or damped by nonlinear transformations in the fire weather model (Erickson et al., 2018). Thus, for example, a dry bias in the model in a certain region will lead to the persistent prediction of higher fire danger values compared to what would be calculated using local observations.

Handling random errors in weather forecasts is traditionally done through the use of ensemble prediction systems where several simulations are performed starting from slightly different initial conditions and model configurations (Molteni et al., 1996; Buizza et al., 1999). Given the expenses of running an ensemble system these simulations are usually conducted at a lower resolution than a single deterministic run. The forecast is then interpreted as probabilistic rather than deterministic. While it has been shown that the probabilistic information contained in an ensemble prediction system might be difficult to interpret

for end-users (Pappenberger et al., 2013), ensembles can boost confidence in the decision process during emergency situations as a cost-loss analysis can be associated to the different scenarios (Cloke et al., 2017). Moreover, ensemble predictions can have more information value than the single deterministic simulation (Richardson, 2000; Zhu et al., 2002). Systematic biases, on the other hand, can be reduced by model improvements. For instance, appropriate post-processing (bias correction) of the atmospheric model (Piani et al., 2010; Di Giuseppe et al., 2013a, b) or post-processing of the sectoral application outputs

(Raftery et al., 2005) can correct resolved processes and improve the final forecast skill.

Given the above considerations, in this paper we assess the performance of the fire danger forecasting system developed for the Copernicus Emergency Management Service by the European Centre for Medium-range Weather Forecasts (ECMWF)

to predict the FWI values where a comparison is performed against observed weather conditions. The system is also assessed in terms of its capability to mark high danger when an event actually occurred looking at the probability of detection of fire during one year of operation in 2017. As the Fire Weather Index is the main index of this system we will concentrate on this model component.

## 2 Methods

### 2.1 FWI calculation

#### 2.1.1 General concept

The Fire Weather Index system provides an indication of fire danger conditions as influenced by four weather parameter, temperature, relative humidity, precipitation and wind speed (Van Wagner et al., 1987). It models the moisture content of dead woody debris of different diameter classes laying on three fuel beds and from these an indication of what would be the rate of fire spread and the fuel available for combustion, It also provides a general indicator of fire danger, the Fire Weather Index (FWI).

A comprehensive description of the FWI system, the interaction between the various components and how these are used in fire management can be found in (Van Wagner et al., 1987; Wotton, 2009). Abatzoglou et al. (2018) showed that FWI exhibits strong correlative relationships to burned area across some non-arid eco-regions globally albeith with only weaker relationships in climatically drier regions (shrubland) with the larger correlation found in the boreal and evergreen temperate forests of western North America. Also Bowman et al. (2017) highlighted how high FWI values are often associated to the most extreme fire activities recorded using Fire Radiative Power observations. As FWI has been shown to provide a good metric for quantifying fire danger globally, the proposed analysis of forecast skills will concentrate on this index (Di Giuseppe et al., 2016a; de Groot et al., 2007).

#### 2.1.2 FWI forecast

For each day indexes of the FWI rating system are calculated operationally at ECMWF using real-time (RT) forecasts. A full description of the modeling components can be found in Di Giuseppe et al. (2016a). The high resolution (HRES) and the ensemble prediction systems (ENS) provide weather forecasts which extend up to 10 days in the future. The atmospheric forcings have a temporal resolution of 3 hours and a spatial resolution of 9km for the high resolution run and 18 km for the ensemble prediction simulations. While the HRES is a single (deterministic) model integration, the ENS provides 51 realizations from perturbed initial conditions and different model physics (Buizza et al., 1999). These ENS forecasts are used to assess uncertainties in the prediction.

A model integration at any nominal time simulates atmospheric conditions at a different local time, depending on the location. FWI calculations are usually performed at 12 noon local time because the model was calibrated using measurements at 12:00 against fire behavior in the most active window (between 14:00-16:00) (Van Wagner et al., 1987). Therefore to produce

a snapshot at 12 noon local time, a temporal and spatial collage of 24 hours time model simulations is performed. Atmospheric fields are cut into 3-hourly time strips using the closest 3-hour forecast outputs and then concatenated together so that the final field is representative of the conditions around the local noon within the 3 hour resolution available (see Di Giuseppe et al. (2016a) for more details). ECMWF implementation for the FWI is initialised once starting from idealised conditions following Wotton (2009) values. It also does not implement any overwintering meaning that the moisture codes are not reset to zero during cold winter months.

### 2.1.3 FWI reference and benchmark

As many forestry agencies still rely on observed meteorological data to provide fire danger, a first assessment of the quality of forecasted FWI will rely on the comparison with observations. Despite several meteorological observations are available through the Global Telecommunication System (GTS) SYNOP network, only a subgroup of stations have at least 30 days of recordings at local noon during 2017 (spatial coverage is given in Figure 1). Many fire prone regions, such as Australia, would not be covered by this comparison. In order to overcome this limitation, a reference dataset of FWI modelled values is also used. This dataset is publicly available through the Copernicus Climate data Store and is constructed using ERA5 reanalysis dataset. ERA5 is the latest of ECMWF reanalysis products which was released at the beginning of 2019. It replaces the previous ERA-Interim database (Dee et al., 2011; Vitolo et al., 2019) providing a much improved spatial resolution and an extensive increment of assimilated observations. Simulations begin in 1979 and are updated in quasi real time with less than a week delay. Fields have a spatial resolution of about 30 km and hourly time resolution. Outputs from ERA5 undergo the same temporal interpolation described in the previous section to provide the model with a composite fire reanalysis product at 12:00 local time. It has to be noted that, compared to local observations, a reanalysis provides a dynamically consistent estimate of the climate state at each time step and can, to a large extent, be considered a good proxy for observed meteorological conditions. Moreover, by combining different observations, reanalysis datasets extend well beyond the natural life of single observational networks and they can provide a more homogeneous spatial coverage than using local observations. From ERA5 we also derive a climatological benchmark simulation (called CLIM hereafter). At pixel level and for every day of the year, CLIM is constructed using 51 randomly sampled values (with replacements) from observed meteorological forcing in the period 1980–2019, excluding the verifying year (2017). CLIM has the advantage of having the same climatology of ERA5, but has no expected predictive skill. The advantage of CLIM is that in theory it has near-perfect reliability with regards to the ERA5 runs since it is produced with the same unbiased forcing data. It should, therefore, score better or equal to the forecast as predictor on time ranges beyond their respective limits of predictability. CLIM is therefore used in this study as a benchmark to rank the expected improvements provided by a forecasting system. A full validation of the FWI database derived from ERA5 can be found in Vitolo et al. (2020)

### 2.2 Observed fire events

While national inventories of wildfire activities exist in many countries, they can be heterogeneous and lack the temporal span desirable for the validation of a fire danger system at the global scale. Satellite observations can supply a valid alternative

especially as they cover remote areas where in-situ observations are sparse (Flannigan and Haar, 1986; Giglio et al., 2003; Schroeder et al., 2008). Daily maps of fire radiative power (FRP) (Kaufman et al., 2003; Wooster et al., 2005) are available from ECMWF since 2003 through the Global Fire Assimilation System (GFAS) (Kaiser et al., 2012; Di Giuseppe et al., 2017,

2018). This dataset has been developed in the framework of the Copernicus Atmosphere Monitoring Services (CAMS) and uses observations from the MODIS sensors on board of Terra and Aqua platforms and assumptions on fire evolution to calculate a continuous record of active fires. The GFAS dataset integrates all available FRP observations available in a day over a regular $0.1 \deg$ grid. According to Wooster et al. (2005), this provides an indication of the cumulative dry mass available for burning which can be then put into a relationship with fire emissions. In this paper, the FRP products are only used as an observations of

fire events. However, FRP values are ignored and only used to derive a mask of fire occurrence based on a minimum detection criteria: $FRP > 0.5 W m^{-2}$ (Kaiser et al., 2012). A "hit" is recorded if the fire forecast predicts fire danger above the 90[th] percentile of its historical values (provided by the ERA5 simulations) when a fire really occurred.

## 2.3    Score metrics

The performance of the the fire forecasting systems to reproduce observed FWI values is assessed using deterministic and

probabilistic scores. Both the synop database and ERA5 are treated as a proxy for observations in the evaluation. To asses the quality of the forecats we use traditional deterministic skill scores such as the mean bias (MB) and the mean absolute error (MAE). For a probabilistic assessment, the continuous ranked probability score is also employed (CRPS; Hersbach (2000)). These metrics are defined as:

$$MB = \sum_{p=1}^{cases} [F_{HRES} - O]$$

$$MAE = \frac{1}{cases} \sum_{p=1}^{cases} \sqrt{[(F_{HRES} - O)^2]]}$$

$$CRPS = \frac{1}{cases} \sum_{t=1}^{cases} \int_{-\inf}^{+\inf} [F_n - O)^2] \, dn$$

where F is the forecast at time step t of N number of forecasts and O is the observed value. While the MB and MAE are applied to a single forecast, the high resolution forecast HRES, the CRPS takes into account the whole distribution of possible values predicted by the ensemble. The CRPS is the continuous extension of the ranked probability score, where $F_n$

is the cumulative distribution function of the predicted ensemble values. Then, the CRPS compares the cumulative probability distribution of the FWI forecast by the ensemble system to the observation. In this sense the CRPS is sensitive to the mean forecast biases as well as the spread of the ensemble (Hersbach, 2000).

While conventional skills score can be employed to assess the quality of the FWI computation, the verification of the FWI as a fire indicator is instead extremely challenging. First, as widely explained, FWI is not a physical measure of fire activity

but of its potential danger, if one were ignited. Therefore high fire danger, while being correctly forecasted, might not result

in active fires if there is no ignition andor aggressive fire suppression. From the verification point of view this means that the identification of false alarms is not meaningful and the verification should mainly rely on hits and misses. Secondly, fires are rare events and, as for any other infrequent phenomena, the verification statistics are heavily influenced by the small number of hits when compared to the total. Still, when the cost of a missed event is high, for example in terms of human lives, the deliberate over-forecasting may be justified (Richardson, 2000; Cloke et al., 2017).

In these cases a positively oriented score such as "hit rate" may be useful especially if related to the case of not having a forecast at all. Also forecast quality does not always equal forecast value (Richardson, 2000). A forecast has high quality if it predicts the observed conditions well according to some objective or subjective criteria. It has value if it helps the user to make a better decision in terms of protective actions (Cloke et al., 2017). For example predicting high temperature and low precipitation in desert areas might be accurate but carries low information content and therefore limited value. Following these arguments and to gain an appreciation of the potential value of the forecasting system globally we use as a metric the Probability of Detection (POD), which measures the fraction of the observed events that were correctly forecast ($POD = hits/(hits + misses)$). Therefore, POD only takes into account observed fires and, unlike other skill scores such as the Brier score, does not suffer from the artificial vanishing due to the high number of correct negative and false alarms (see Stephenson et al. (2008); Ferro and Stephenson (2011) for a discussion on this problem).

## 2.4 Fire regions

The global assessment of the fire forecast skills is mostly provided as an average over selected regions even if the calculation of the various scores is performed at pixel level by interpolating the model grid over the verification points. For an assessment at the continental scale, we use the fire macro-regions defined by the Global Fire Emission Database, GFED4 (Giglio et al., 2013). These macro-regions are characterized by different fire regimes and are very roughly homogeneous in their burning emissions contribution (Giglio et al., 2013). Inside these regions we also select 3 areas at national/regional level - California, Portugal and Chile - which experience recurrent intense fire episodes and saw major events taking place in 2017 (Figure 1). Events in these locations are also analyzed in detail.

## 3 Results

### 3.1 Skill in the FWI prediction

The first assessment looks at the capability of ECMWF fire forecast to reproduce the same FWI values as would be estimated from the network of local stations but up to 10 days ahead. The selected stations (Figure 1), which have at least 30 records during 2017 at local noon, are used to perform an analysis of MB and MAE at different lead times (Figure 2). For comparison also FWI calculations using ERA5 are included, which provides a validation of the assumption that ERA5 is a good proxy for observations. As expected there is a performance degradation going towards longer lead times however the increase is in within the distribution and mean biases are limited to few units even at day 10. However caution is in order, as depending on

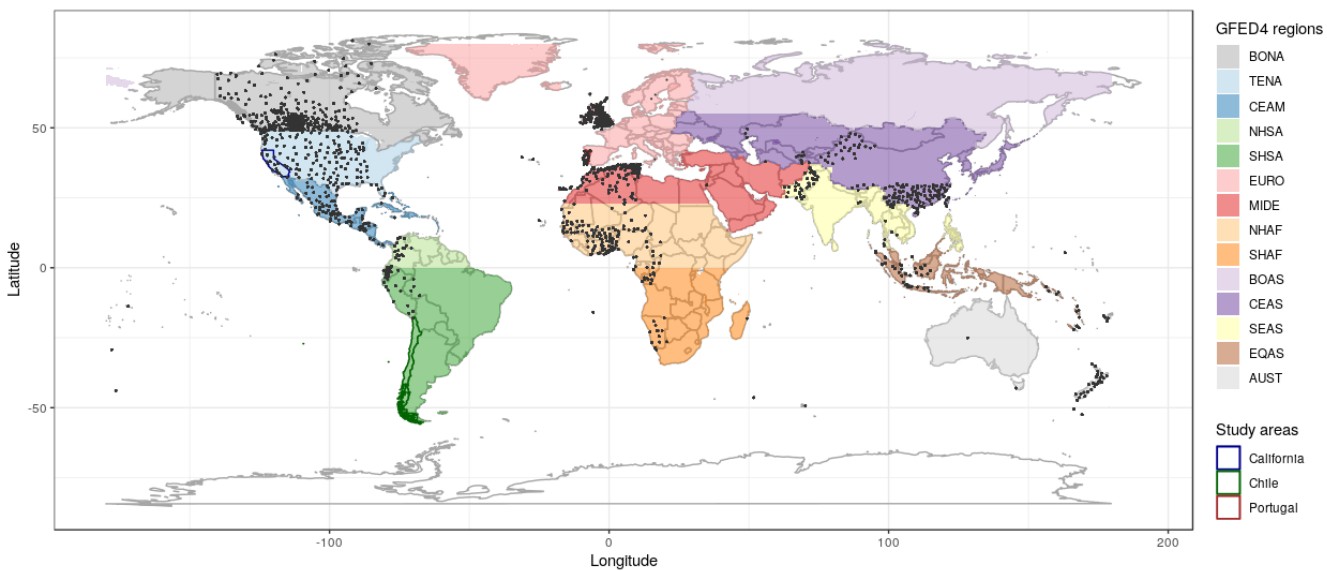

**Figure 1.** GFED4 regional classification and the 3 countries selected to showcase the fire forecast performances (California, Chile, and Portugal). The black dots show the spatial distribution of weather stations from the synop network which have at least 30 observations recorded at local noon in 2017.

the calibration procedure adopted, few units could mean a mismatch in danger level classification. The mean absolute error provides information on the residual amplitudes. FWI from reanalysis have the sharpest skills, as expected, while the mean absolute error rapidly increases with lead times. However the distribution of MAE values clearly shows that in selected events the discrepancies between observed and predicted values is confined to few units even 10 days ahead. As it is recognised that in some regions in the tropical areas the number of stations is very reduced a similar analysis is also performed using ERA5 as the verifying databases (see figure 5 in the following section), which however confirms the general conclusions.

Despite its importance the analysis performed using the synop network is point-wise and does not homogeneously cover all the regions where fires are relevant. Moreover, MB and MAE are based on high resolution forecasts and do not provide information about the performance of the ensemble forecasting system as a whole. A global assessment of the performances of the system is provided by the comparison between the CRPS curves for the forecast and CLIM when both are scored against ERA5 in 2017 (Figure 3). The CRPS calculated from the CLIM database provided a useful benchmark for the forecast as it defines the error above which the information content stored in the forecast would be equivalent to the information provided by the climate. The first interesting information from comparing the two experiments is how far in advance there is skill in predicting fire danger from weather forecast. In fact the interception between the CRPS curve from the forecast run and the CLIM run marks the overall length of the predictability windows, i.e where the system still provides skills above

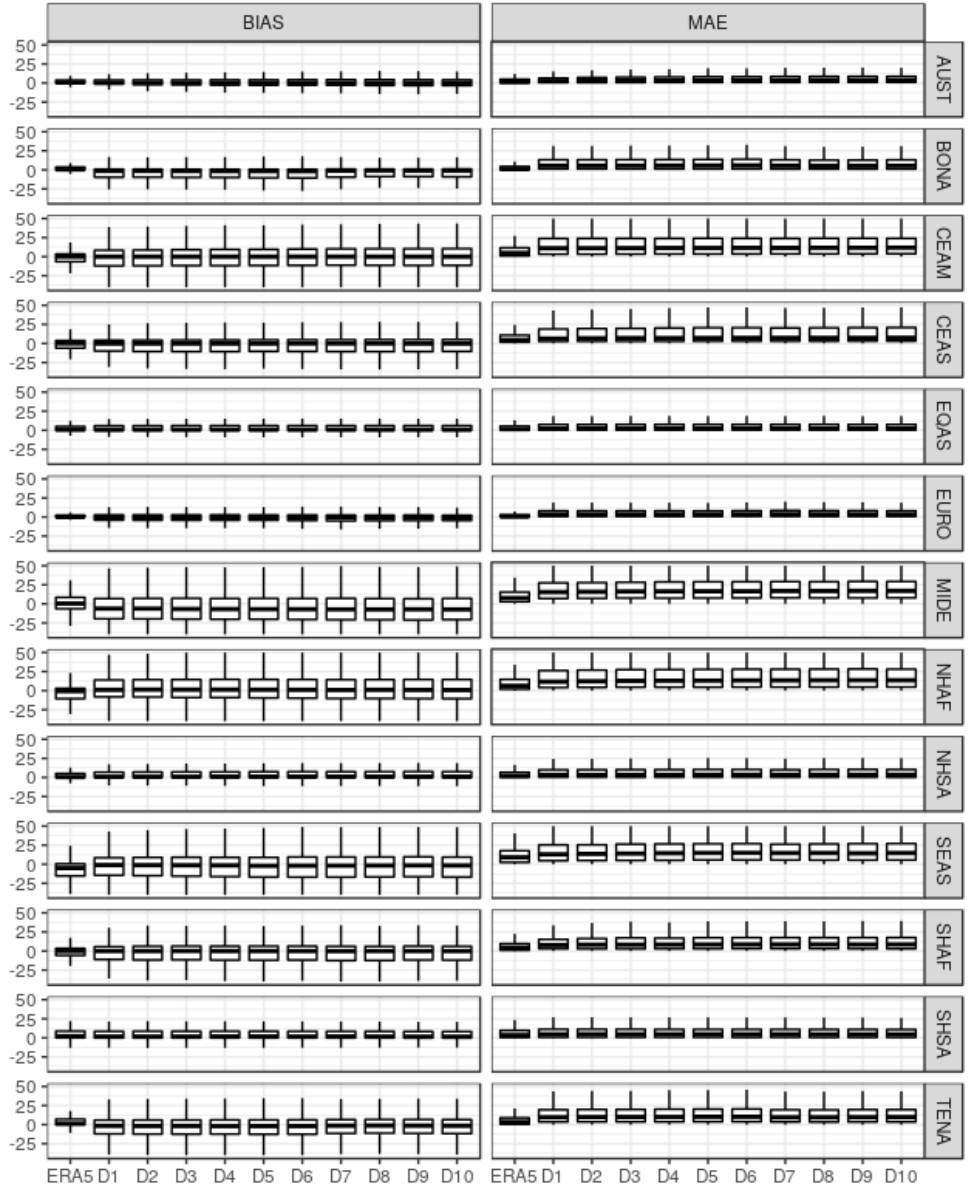

**Figure 2.** Comparison between modelled and observed FWI value across the GFED macro-regions. FWI calculated using ECMWF high resolution forecasts at different lead times are verified against ERA5 simulations. The box plots describe the distribution of values across the observation points for one year of simulations in 2017. Mean bias is plotted in the left panel and mean absolute error in the right panel.

climatology. Encouragingly if we look at the global average, the window of predictability is longer than the 10 days range provided here, which also suggests that there is scope for extending the prediction to the sub-seasonal and seasonal time scales. The discontinuity visible at day 6 is an artefact due to the change in temporal resolution in the ecmwf forecast. Up to day 6 forecasts are stored 3 hourly and only 6 hourly after this time step.

There are some regional differences in the skill provided by the ensemble forecast. Regions covered by Boreal forests (e.g. BOAS, BONA, part of CEAS) have the largest predictability with the maximum gaps between the forecast and the climate CRPS scores (Figure 4). Savannah regions (NHAF, AUST,SHAF) tend to have a shorter window of predictability with the forecast CRPS curve approaching at a shorter lead time than the CLIM ones. The regional differences in the prediction of the forecast FWI when compared to ERA5 derived databases are related to the skills of the forecast which then project in the accuracy in the FWI simulation. While temperature predictions skills are globally mostly uniform, a complex picture emerges for the forecast skills of precipitation in all global models used for numerical weather prediction including ECMWF model. Prediction of precipitation in the mid latitudes is notoriously more accurate than in the tropics due to the connection with frontal systems driven by large scale dynamics (Simmons and Hollingsworth, 2002). Convective precipitation which is the main source of rainfall in the tropics is by nature stochastically occurring and therefore more challenging to predcit. Although the gap has been filled through the years forecast predictions in the Southern extra-tropical region is less accurate than the equivalent in the Northern hemisphere due to the availability of a better observing system to constraint the forecast initial conditions (Haiden et al., 2019). These considerations could largely explain the better performances of the FWI predictions in the northern hemisphere for the year taken in consideration. However it has to be noted that forecast skills have strong year to year variations with expected increased skills in the tropic when large scale phenomena such as the Madden-Julian Oscillation (MJO) and/or the El Niño Southern Oscillation (ENSO) take place. Under these phenomena the predictability of the tropics and of extra-tropical regions can substantially improve through teleconnections (Vitart, 2014).

Exceptionally poor is the performance in the two South American regions where the forecast at any lead time is below the climate line. As mentioned CRPS is heavily influenced by the forecast bias which can induce a fast decline in the CRPS curve. Looking at the mean bias as a function of the lead time (figure 5) it is evident how these two regions are indeed strongly affected by systematic biases with the largest values recorded at least in the first three days of forecast. In general for all the regions the decline in CRPSS (Fig 5 ) can, to some extent, be explained by the negative bias (too low FWI values when compared to ERA5-FWI). Interestingly the bias of the forecast is not spatially consistent, it is generally larger in the Southerns Hemisphere regions and lower in the Northern Emisphere, in agreement with what discussed on the expected skills of the weather forecast. The consistent negative bias at all lead times also highlights that there is scope to improve the overall skill of the prediction through bias corrections of the meteorological forcing a (Piani et al., 2010; Di Giuseppe et al., 2013a, b).

As a general conclusion and provided the possible year to year variability in skills, the general picture that emerges is that for most of the areas weather forecast provides predictive skills for the FWI beyond 10 days.

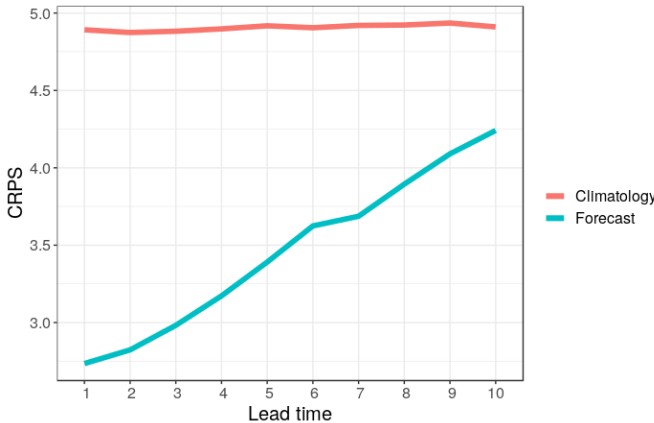

**Figure 3.** CRPS for the ensemble fire danger forecast (blue line) and the CLIM database constructed using a random selection of ERA5 years not including the verifying year (red line). Data have been globally aggregated and the forecast is available up to day 10 horizon.

## 3.2 Skill in detecting fire events

Being able to predict the observed value of FWI does not equal to being able to pinpoint occurred fires. Figure 6 shows the location of recorded fires in 2017 based of Fire Radiative Power (FRP) observations from MODIS sensors as integrated by the GFAS platform (Kaiser et al., 2012; Di Giuseppe et al., 2016a, b). Fires covered large parts of the globe in 2017, with 157,631 cells recording $FRP > 0.5Wm^{-2}$. To understand the capability of the FWI to match the occurrence of actual fires we assume that an active fire is correctly predicted if the FWI is greater than the $90^{th}$ percentile of its distribution of values
here defined using the ERA5 database. Figure 7 shows a summary table of the mean probability of detection (POD) by region for all events in 2017 at forecast day 1 to 10. Given the intrinsic limitations of the POD as skill metric, CLIM could provides a useful benchmark to understand the incremental skill provided by the forecast. The POD provided by CLIM was found below 0.1 in all regions and is therefore not shown in the table.

CLIM has no skill in predicting fire events as it always provides the lowest POD, corresponding to a probability of detection
below 10%, even when compared to day 10 forecasts. On the other side, forecasts POD vary widely by region, with Europe (EURO) and Boreal North America (BONA) being the only regions with POD above 0.5. These are mostly temperate regions where vegetation is dominated by forests and fuel is abundant and where fire danger is moisture limited. In these regions the FWI is a good predictor of fire danger (Di Giuseppe et al., 2016a). It has to be noted that the FWI does not take into account management measures that could introduce a relevant number of "false-alarms". Central America, the Middle East and the
northern hemisphere areas, Africa are characterized by a POD in the range 0.2-0.5 as in most of the tropics, where fires usually occur in grass-shrub lands. Here fuel is scarce and weather plays a less relevant controlling role. Also it has to be noted that the statistics here are likely to be contaminated by many agricultural and prescribed fires that are considered 'events' and which would dilute some of the skill in regions where annual cropland is high or are heavily managed.

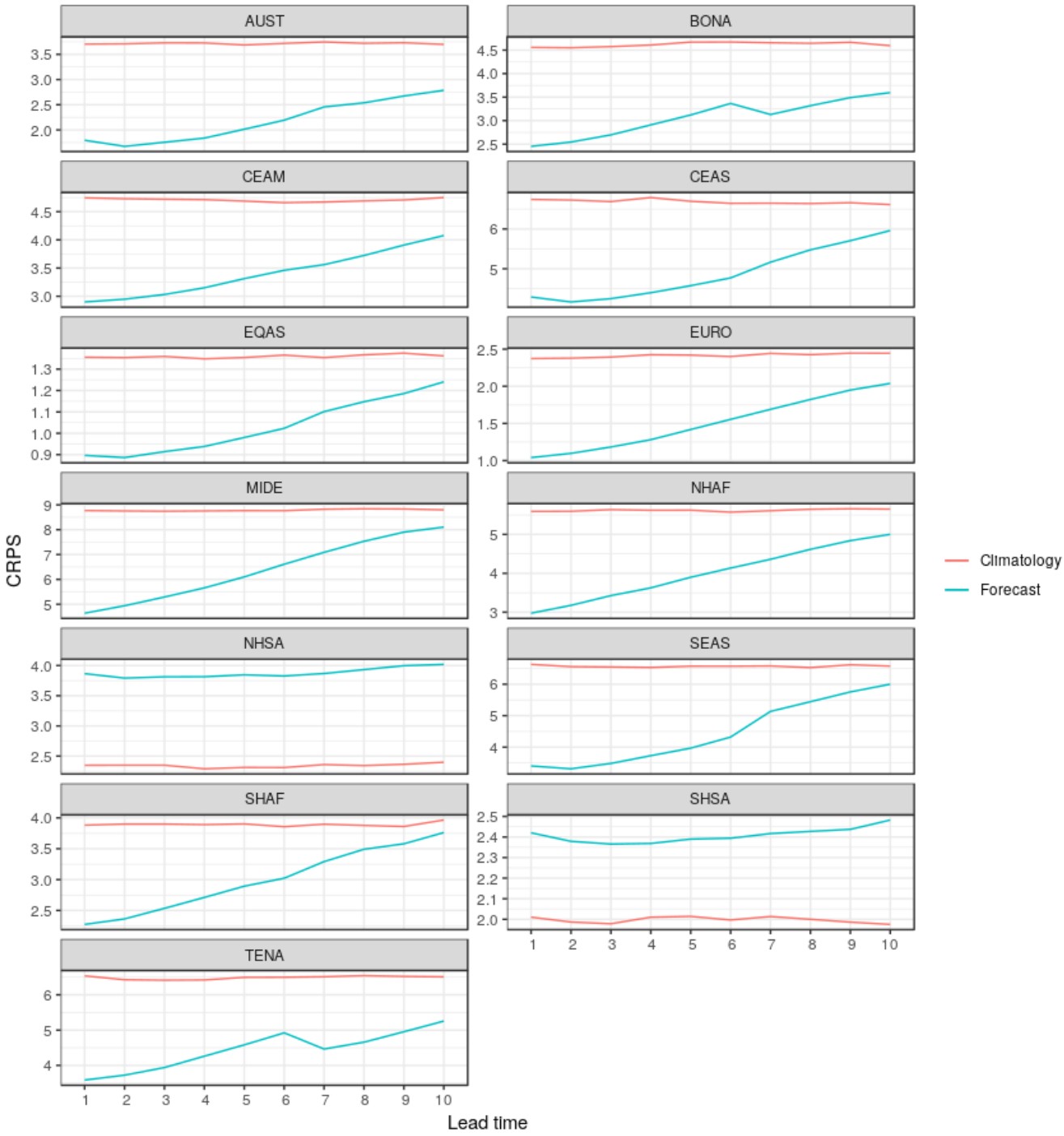

**Figure 4.** As figure 3 but with aggregation performed on the GFED macro-regions

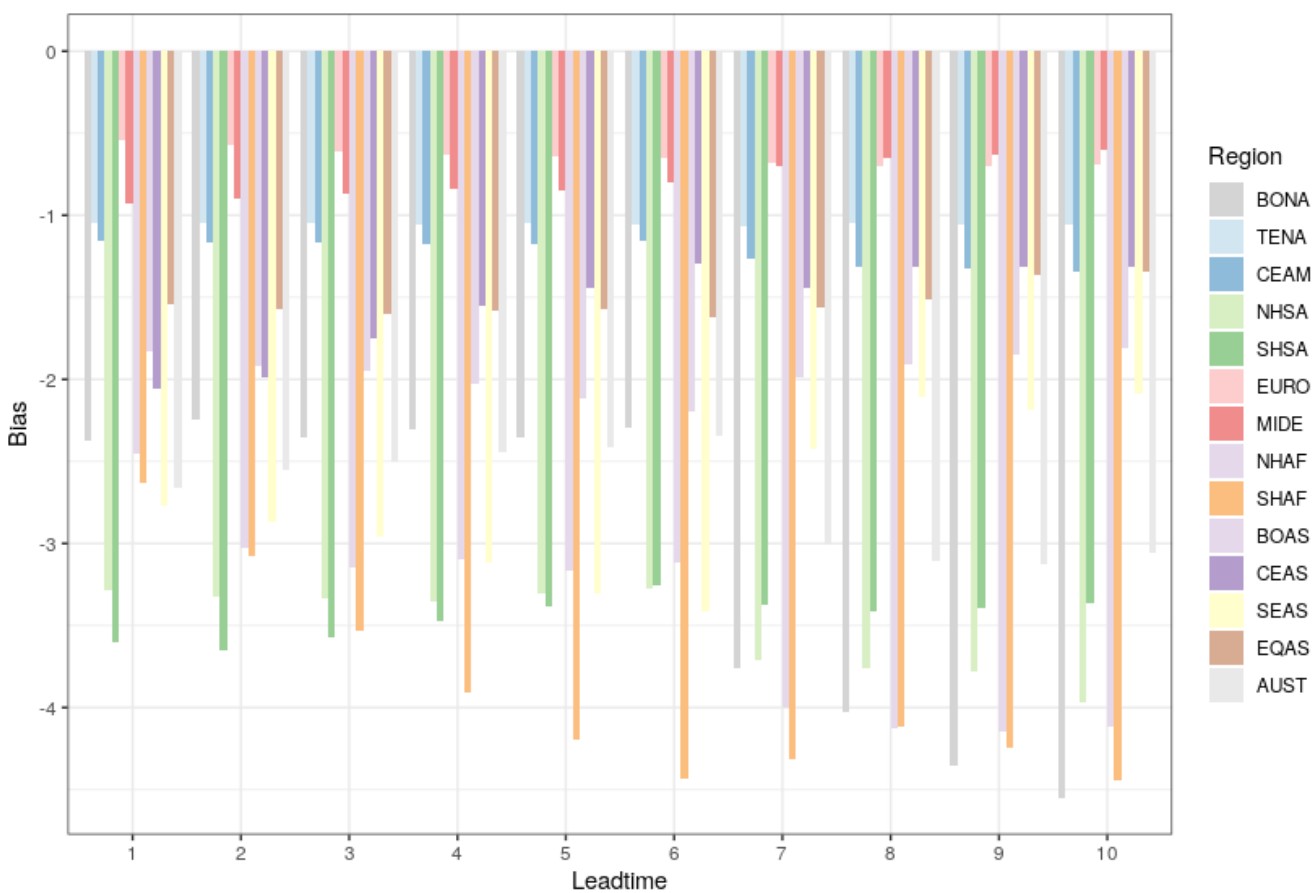

**Figure 5.** Mean forecast bias as a function of lead times and clustered by GFED macro-regions. The forecast bias for 2017 is assessed using the ensemble mean against ERA5 based FWI database.

One important exception is the very low performance of the fire forecast in Equatorial Asia (EQAS) and South East Asia
(SEAS) where the system seems to have a predictability below 0.2 (only 20 % of fires corresponded to FWI above the 90th percentile). de Groot et al. (2007) highlighted how FWI is not the best indicator in this areas and a fire early warning system should mostly rely on the drought code. There are a number of factors that could contribute to this low usability of the FWI in these areas. Fires in these regions are mainly caused by humans for the purposes of cleaning the land for establishing plantations (Field et al., 2009; Benedetti et al., 2016) and weather, which is the only driver of the FWI, is not the main fire
trigger. However it has to be noted that 2017 was a very wet year in EQAS and anomalously low FWI were predicted (see for exemple Figure 7 in Vitolo et al. (2020)) with a consistent low emissions recorded by GFED. The low level for fire activities in 2017 means that the applicability of the results for this region in 2017 might not extend to other years with stronger activities. Also Australia (AUST) has a very low POD for the FWI possibly being a fuel limited ecosystem (Krawchuk et al., 2009). The

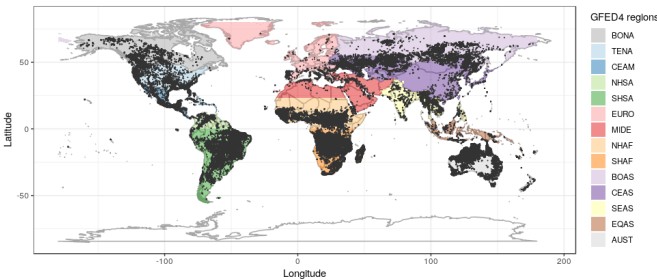

**Figure 6.** GFED4 micro-regions with superimposed the locations of all the 2017 events reported as active fires using the Fire Radiative Power (FRP) observations from MODIS ingested into the GFAS system.

main picture that emerges is that while weather forecast can provide skilful prediction for FWI at least 10 days ahead, this fire danger index has in many areas a scarce capability to pinpoint emerging fires.

### 3.3 2017 case studies

Figure 7 provides an averaged assessment of the global performances of the FWI to mark any fire pixels identified during 2017. This global statistic includes small fires and events that are not exclusively driven by weather conditions. FWI skill could improve locally, especial when important fire events are considered. It is important to understand how the information provided by a 10 day forecast could be used in real cases when the information is intended to aid emergency responses. Here we will analyse three cases of fire events that took place in 2017, which proved to be an year with extreme fire episodes across the globe. The 2017 wildfire season involved wildfires on multiple continents and also, possibly unprecedented events when melted peat bogs ignited in Greenland. The year 2017 started with an extended fire in central Chile that lasted almost all of January. Strong winds, high temperatures and long-term drought conditions led to an event that has been described as the worst wildfire in Chilean history (Bowman et al., 2018). Fires in the central regions of O'Higgins, Maule and Bío Bío south of Santiago were difficult to control. Although fire activities where recorded since July 2016 they became particularly intense in January 2017. In June, between day 17 and 18, another devastating fire hit Portugal. It claimed more than 60 lives mostly recorded in the Pedrógão Grande area, 50 km southeast of Coimbra. A persistent heatwave had been building in the region, with temperatures above 40C, which are highly unusual for the season. Moreover, relative humidity levels below 30% had a role to the intensification of the deflagration and the spread of the wildfire, which raged out of control for several days (Boer et al., 2017). Finally in October, extensive wildfires raced just north of the San Francisco Bay Area in California causing historic levels of death and destruction. These named 'Wine Country' wildfires were the most destructive in California history, with 44

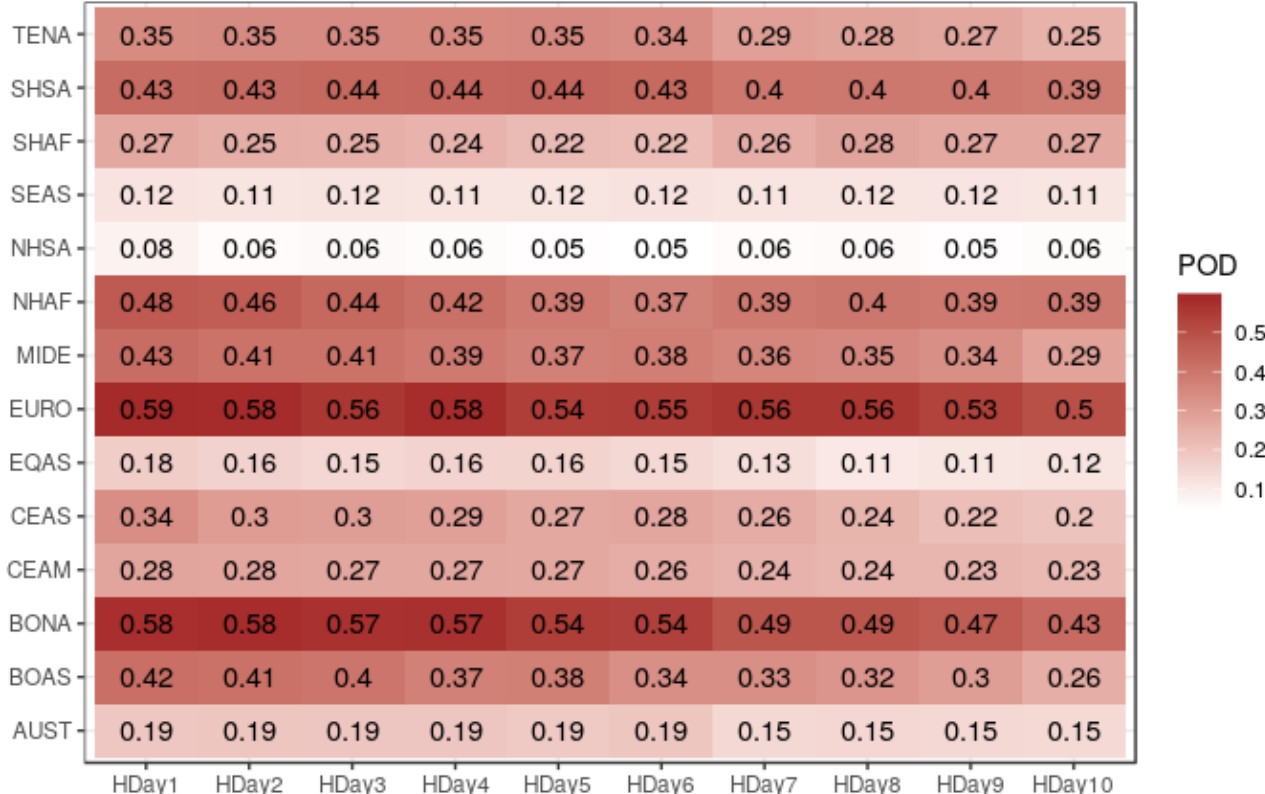

**Figure 7.** Regional Probability of Detection (POD) for the high resolution forecasts from day 1 to 10 (HDAY1/10). Events where FRP $\geq 0.5$ $Wm^{-2}$ are categorized as "hits" and compared to FWI prediction above the high warning level (90th percentile of climatology). The statistics are constructed using FRP observations detected in 2017.

deaths; the loss of 9,000 buildings; damage to approximately 21,000 structures; $10 billion of insured losses; and substantially greater total economic loss (Nauslar et al., 2018; Mass and Ovens, 2019).

Figures 8 show the information that could have been provided for the study areas by the 10-day fire danger high resolution forecasts (HRES), had these been already available. Each plot shows on the x-axis the dates in which FRP was observed and, on the y-axis, the dates forecasts were issued. The cell in the bottom left corner shows the percentage of pixels in the study area that are expected to be above the $90^{th}$ percentile of the FWI climatology for that pixel and day of the year. The forecasts for day 2 to day 10 are on the same row. The forecasts issued on the following day are one row above and so forth. The dashed lines show the observed Fire Radiative Power (see also secondary y-axis).

The reader is reminded that active fires are triggered by highly unpredictable events (ignition) which are not accounted for in the FWI system. The FWI is not supposed to provide the exact localization of the event but an indication of potential fire activity. Large areas can be affected by anomalous conditions in the proximity of where the event really occurred. However it

**Table 1.** Events summary table.

| Country | Region | Start date | End date | Main event | Location |
|---|---|---|---|---|---|
| Chile | O'Higgins, Maule, Bío Bío | 01-01-2017 | 31-01-2017 | 26-01-2017 | 36° 46'S; 73° 03'W |
| Portugal | Pedrogao Grande | 01-06-2017 | 30-06-2017 | 18-06-2017 | 39° 55'N ; 8° 08' W |
| USA | California | 21-09-2017 | 20-10-2017 | 09-10-2017 | 38° 34'N; 122° 34' W |

is encouraging that there is some capability for the forecast to detect the increase in fire danger associated to the three events even if with different intensities and sharpness. For the Chile case, for example from mid-January often around 70% of the area exceeded the high danger threshold. The FRP spike occurred on the 26th of January and while the forecast was not able to capture this increase in fire activity, looking at the whole monthly sequence there is an indication of increased danger conditions even at 10 days lead time. However it is recognised that the signal extends for long time and does not mark the peak of the fire activities. A much better timing of the event was instead forecast during the Portugal and California fires which were very well predicted 10 days ahead.

## 4   Conclusions

In the last years, ECMWF has been involved in the EFFIS development by providing weather forcing and fire danger calculations using its medium-range weather forecasts. Global fields of FWI are calculated daily using the high-resolution (9 km) forecast up to 10 days ahead. The 18 km resolution ensemble prediction system provides additional 51 realizations based on slightly different initial conditions and/or using different model configurations (Molteni et al., 1996). These datasets are freely available in line with the data and information policy of the Copernicus program which intends to provide users with free, full and open access to environmental data. Using one year of pre-operational service in 2017 we have showcased the potential of the use of weather forecasts to support the monitoring of fire danger conditions and planning in case of a potential emergency. Weather forecast provides skillful information to derive FWI values up to 10 days ahead. Looking at the Continous Ranked Probability Score for the forecast in comparison to climatological simulations it was shown that predictive skills could extend also beyond the provided forecast range for most of the GFED macroregions. Similarly to other sectoral applications (Wetterhall and Di Giuseppe, 2017) there is scope to extend the prediction to the sub-seasonal and seasonal time frame (S2S). On the other hand a good skill in forecasting FWI values did not translate into a satisfactory probability of detection for real fire events. When all observed fires in 2017 where matched to high values of FWI ($> 90th$ percentile) only the Boreal regions for which the FWI has been calibrated had a POD above 50%. Mid and high latitude forested areas, where fuel is abundant have the highest predictability while in savana/shrub-land regions the relationship between FWI and fire occurrence weakens. Still global statistics are likely to be contaminated by many agricultural and prescribed fires that are considered 'events' and which could dilute some of the skill in regions where annual cropland is high or are heavily managed.

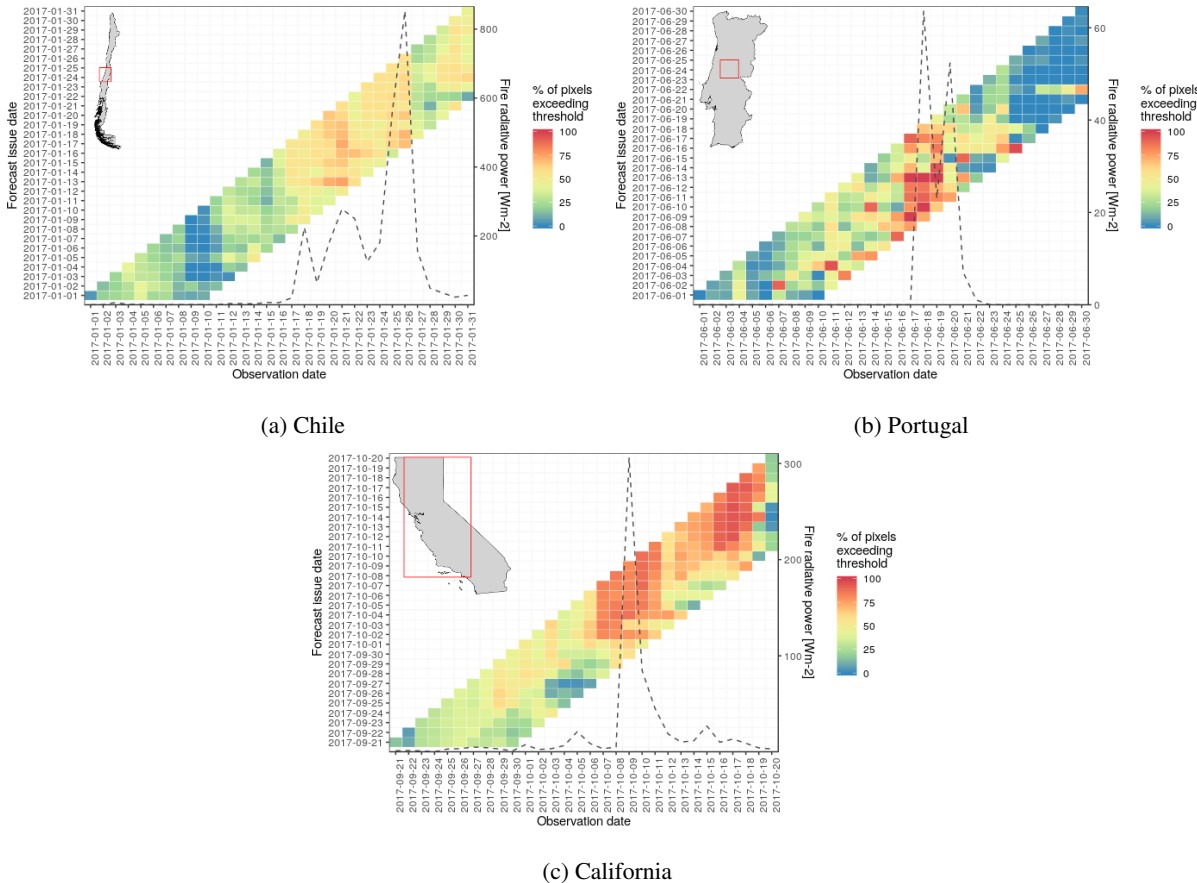

(a) Chile

(b) Portugal

(c) California

**Figure 8.** Comparison of Fire Radiative Power (gray dashed line with axis on the right hand side) with FWI forecasted using the deterministic high resolution model for: (a) Chile, (b) Portugal and (c) California. FWI is color coded based on the percentage of pixels exceeding the high danger level calculated at the country/state level. Each of the panel refers to a specific fire event described in the text and the statistics have been calculated over the red boxes.

Looking at large fire events in Chile, Portugal and California which occurred in 2017 we have shown that there are regional differences and in Portugal and California the forecast was accurate up to 10 days ahead. Another interesting aspect attached to the use of weather forecasts is the use of probabilistic information. The quantification of forecast uncertainties through the use of ensemble predictions is something still pretty new in fire forecasting. However it opens great opportunities in terms of adding a confidence level to the the fire prediction. These aspects will be investigated in follow on work.

**Code availability**

In the spirit of reproducibility, function and workflow to generate the results of this manuscript are available on public repositories (Vitolo et al., 2018; Vitolo and Di Giuseppe, 2020).

*Author contributions.* FDG designed the experiments and wrote the paper, CV performed the verification analysis, BK, CB and PM run the experiments and contributed to the creation of the ECMWF operational system, JS-M contributed to the design of the experiments. All authors revised the paper

325 *Competing interests.* The authors declare no competing interests

*Acknowledgements.* This work was founded by the EU Project ANYWHERE (Contract 700099) and the Global Fire Contract 389730 between the Joint Research Centre and ECMWF.

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
