# Peer review of "Fire weather index: the skill provided by ECMWF ensemble prediction system"

_Natural Hazards and Earth System Sciences, 2020_

## Referee Comment (RC1) · Anonymous Referee #1 · 23 Mar 2020

The authors present an assessment on the skill of the GEFF system in forecasting fire danger up to 10 days ahead. The system uses as proxy for fire danger the FWI index predicted from the ECMWF 10-day forecasts. The authors use FWI computed from a number of SYNOP observations as well as the ERA5 reanalysis as a substitute for weather observations.

It is my opinion that this work is important as it well documents the system and in general highlights its strengths and weaknesses in terms of deterministic as well as probabilistic forecasts. The use of the "standard" GFED4 regions for comparison is a good idea, even if they encompass large regions with heterogeneous fire regimes.

I recommend it for publication with minor reviews pending the following suggestions are addressed.

- p. 3 section 2.2.1 - the text is a direct copy of other works, should be summarized with a reference to the original work

- p. 4 line 107 - using the era5-base reanalysis without proper validation is a bit problematic. Key findings should be summarized here.

-section 3.1 (last paragraph) - The system is worse than climatology for the 2 South American regions, authors omit this and do not postulate any reason for this poor performance

-section 3.1 - due to a poor spatial coverage of SYNOP stations in tropical areas, I suggest the analysis be done with ERA5 data as a proxy for SYNOP, and results provided in supp. material

Some general remarks and comments:

- p. 2 line 40 - authors ignore the uncertainty in initial state which can also lead to forecast error

- p. 2 line 57 - authors should mention that the configuration used for the ensemble forecasts is done at lower spatial resolution

- p. 5. line 134 - I would suggest replacing "era5 simulations" with "era5 reconstructions" here and elsewhere.

- p. 5. line 138 - replace "quality of the computation" with "quality of the forecast"

- p. 9 line 199 - replace "Boreas" with "Boreal" (also in the last paragraph of the text.

- p. 13 line 265 - replace "signal extend" with "signal extends"

- p. 15 line 277 - replace "predictive skills" with "predictive skill"

- p. 15 line 277 - add "for most of the GFED4 regions studied" after "predictive skill"

- p. 15 line 278 - fix reference format

- p. 15 line 283 - also the FWI was developed for Boreal forests which might explain its better performance in those regions

---

## Referee Comment (RC2) · Anonymous Referee #2 · 22 Apr 2020

Di Giuseppe et al. provide an assessment of 10-day Fire Weather Index for different regions of the globe and for three fire events in Mediterranean fire environments, using one year's worth of data. Both the skill of FWI forecasts are evaluated relative climatology, as is the probability of detection for fire events (characterized by FRP) across the standard GFED fire regions.

The paper provides a useful reference for the ECMWF/JRC's fire weather products, is well-written and for the most part is technically sound. Specific comments are listed below. The main requirements prior to publication are 1) additional discussion of why the forecast performance varies in different regions in terms of the underlying performance of the weather inputs 2) additional context for the 2017 fire season in different parts of the world.

[Figure]

****Specific comments**** L15: suggest changing 'forestry agencies' to 'fire management agencies'

L76: Plese list the four weather inputs to the FWI system, and also describe how the ECWMF initializes the moisture codes in the spring and shuts them down in winter in seasonally cold regions (an example for the Canadian Wildfire Information System is described here: https://cwfis.cfs.nrcan.gc.ca/background/dsm/fwi).

L80: Abatzoglou et al. (2018) showed that the FWI is correlated with burned area across some, rather than most, non-arid regions. Please list here which regions those were.

L96: by 'time strips', do you mean 'time steps'?

L105: suggest replacing 'substitutes' with 'replaces'.

L111: as far as reanalysis being a good proxy for meteorological observations, please mention the extent to which is the case for the ECMWF products, drawing on Vitolo et al. (2019) for ERA-I, and how this might have changes for ERA5.

L153: suggest changing 'if there is no ignition' to 'if there is no ignition and/or aggressive fire suppression'.

L180: throughout, the BOAS region is missing from the analysis. Was there a reason for this, or should it be added?

L182: As far as performance degradation goes, in Figure 2, the differences between ERA5 and the forecasts can be seen, and the differences between regions can be seen, but the panels in Figure 2 are too small to be able to see any performance degradation with increasing lead time.

L194: In Figure 3, is there a reason for the discontinuity between day-6 and day-7 lead times? Similarly for the more apparent discontinuities in Figure 4 for BONA, SEAS. And for TENA, what is the possible reason why the CRPS is lower at lead 7 than lead

6?

L199: Change 'Boreas' to 'Boreal'.

L199-L204: It is hard to understand how FWI calibration (in the sense of interpreting it as a fire danger metric) influences forecast performance for different regions. Instead, the differences are more likely due to performance in forecasting the temperature, relative humidity, wind speed and precipitation input variables. Here, please describe possible drivers of these differences in terms of known differences forecast skill for the four inputs, which I trust are evaluated routinely at the ECMWF.

L217: BONA is by definition is not a temperate region, please correct.

L220: change 'false-alarm' to 'false-alarms'?

L222: Reference required for the statement of weather playing a less relevant role compared to fuel availability in this region.

L225: the de Groot et al. (2007) study is only relevant to the EQAS region, not SEAS, and despite recommending other FWI subcomponents, does not appear to say that the FWI is not a good indicator in the area – please correct. Furthermore, while fires in this area are related to land clearing, the Abatzoglou et al. (2018) study shows that the FWI is highly correlated with burned area over EQAS at longer time scales, and as strongly as anywhere else in the world. The much more likely reason for the poor performance was that 2017 was a record low fire year over EQAS (see GFED dry matter emissions estimates here: https://www.geo.vu.nl/~gwerf/GFED/GFED4/tables/GFED4.1s_DM.txt), and there was simply too little fire activity to make a meaningful prediction, as it was probably too wet for any serious burning. Please correct this, ideally with a brief mention of whether the FWI was anomalously low relative to the ERA5 FWI climatology.

L239: referring to the GFED emissions tables above, 2017 was not extreme across the globe. Please correct.

---

## Author Comment (AC1) · 27 May 2020

The authors present an assessment on the skill of the GEFF system in forecasting fire danger up to 10 days ahead. The system uses as proxy for fire danger the FWI index predicted from the ECMWF 10-day forecasts. The authors use FWI computed from a number of SYNOP observations as well as the ERA5 reanalysis as a substitute for weather observations. It is my opinion that this work is important as it well documents the system and in general highlights its strengths and weaknesses in terms of deterministic as well as probabilistic forecasts. The use of the "standard" GFED4 regions for comparison is a good idea, even if they encompass large regions with heterogeneous fire regimes. I recommend it for publication with minor reviews pending the following suggestions are addressed.

We would like to thank the reviewer for the overall positive comments and the suggestion that we have tried to address. Detailed answers to the comments can be found in the following

- p. 3 section 2.2.1 - the text is a direct copy of other works, should be summarized with a reference to the original work

Yes it is very similar to the previous paper as it briefly describes the FWI. I have shortened it even further and referred to the previous publication

*The Fire Weather Index system provides an indication of fire danger conditions as influenced by weather \citep{vanwagner:87}. It models the moisture content of dead woody debris of different diameter classes laying on three fuel beds and from these an indication of what would be the rate of fire spread and the fuel available for combustion, It also provides a general indicator of fire danger, the Fire Weather Index (FWI).*

- p. 4 line 107 - using the era5-base reanalysis without proper validation is a bit problematic. Key findings should be summarized here.

This is a good point and era-5 based FWI reanalysis has been fully validated in a paper that is under review in the Scientific Data journal. Considering the timing that paper will be published ahead of this and probably could be fully referenced. For now we have added a reference as submitted as follows

*A full validation of the FWI database derived from ERA5 can be found in \cite{vitolo2020}*

-section 3.1 (last paragraph) - The system is worse than climatology for the 2 South American regions, authors omit this and do not postulate any reason for this poor performance

Sorry for this omission. The very bad results in these regions needed some clarifications. We are unsure as to what degradates the CRPS so much that even at day 1 is worse than climatology. As CRPS is heavily affected by systematic biases one of the reasons could be due to a systematic bias in the weather inputs in these regions that then project into a bad performance on the FWI calculation. Strong surface biases are a combination of land -atmospheric exchange process and clouds. We have provided a new plot here which assess the biases and have written an extensives justification which reads

*Exceptionally poor is the performance in the two South American regions where the forecast at any lead time is below the climate line. As mentioned CRPS is heavily influenced by the forecast bias which can induce a fast decline in the CRPS curve. Looking at the mean bias as a function of the lead time (figure*

*\ref{fig:bias_region}) it is evident how these two regions are indeed strongly affected by systematic biases with the largest values recorded, at least in the first three days of forecast. In general for all the regions the decline in CRPSS (Fig \ref{fig:bias_region} ) can, to some extent, be explained by the negative bias (too low FWI values when compared to ERA5-FWI). Interestingly the bias of the forecast is not spatially consistent, it is generally larger in the Southern Hemisphere regions and lower in the Northern Hemisphere, in agreement with what discussed on the expected skills of the weather forecast. The consistent negative bias at all lead times also highlights that there is scope to improve the overall skill of the prediction through bias corrections of the meteorological forcing a \citep{piani:10,digiuseppe:13a,digiuseppe:03b}.*

-section 3.1 - due to a poor spatial coverage of SYNOP stations in tropical areas, I suggest the analysis be done with ERA5 data as a proxy for SYNOP, and results provided in supp. material

The same figure used for the previous analysis also addresses this question as it provides the comparison with ERA5 on all the points x

Some general remarks and comments:

- p. 2 line 40 - authors ignore the uncertainty in initial state which can also lead to forecast error - p. 2 line 57 - authors should mention that the configuration used for the ensemble forecasts is done at lower spatial resolution

Yes thanks the sentence was a bit convoluted and has been reworded as

*When weather forecasts are used in place of observations, uncertainties can be introduced. Sources of uncertainty can be: (i) the limited knowledge of the initial state and (ii) the misrepresentation of physical processes.*

Also we have mentioned the lower resolution of the ensemble forecast in the following sentence

*Given the expenses of running an ensemble system these simulations are usually conducted at a lower resolution than a single deterministic run.*

- p. 5. line 134 - I would suggest replacing "era5 simulations" with "era5 reconstructions" here and elsewhere.
Simulation is a more used word in the NWP community and we have decided to retain it
- p. 5. line 138 - replace "quality of the computation" with "quality of the forecast"
Word replaced
- p. 9 line 199 - replace "Boreas" with "Boreal" (also in the last paragraph of the text.
Word replaced
- p. 13 line 265 - replace "signal extend" with "signal extends"
Corrected
- p. 15 line 277 - replace "predictive skills" with "predictive skill"
Corrected
- p. 15 line 277 - add "for most of the GFED4 regions studied" after
Added
- p. 15 line 278 - fix reference format
Done
- p. 15 line 283 - also the FWI was developed for Boreal forests which might explain its better performance in those regions

This consideration has been added

---

## Author Comment (AC2) · 27 May 2020

Di Giuseppe et al. provide an assessment of 10-day Fire Weather Index for different regions of the globe and for three fire events in Mediterranean fire environments, using one year's worth of data. Both the skill of FWI forecasts are evaluated relative climatology, as is the probability of detection for fire events (characterized by FRP) across the standard GFED fire regions. The paper provides a useful reference for the ECMWF/JRC's fire weather products, is well-written and for the most part is technically sound. Specific comments are listed below. The main requirements prior to publication are 1) additional discussion of why the forecast performance varies in different regions in terms of the underlying performance of the weather inputs 2) additional context for the 2017 fire season in different parts of the world.

Thanks for the overall positive comments to our work. We have analysed in more detail the result in figure 4 which diversifies FWI skill in different parts of the globe and calculate the bias to put this in context to the performances of the underlying weather forecasts.

(1) We have added a new plot looking at the biases that could explain why the CRPS tends to be larger in tropical regions and South America.
(2) we have added some context for the fire season in 2017, extending the description of the specific case studies

****Specific comments****
L15: suggest changing 'forestry agencies' to 'fire management agencies'

Changed

L76: Please list the four weather inputs to the FWI system, and also describe how the ECWMF initializes the moisture codes in the spring and shuts them down in winter in seasonally cold regions (an example for the Canadian Wildfire Information System is described here: https://cwfis.cfs.nrcan.gc.ca/background/dsm/fwi).

We have now listed the 4 input variables of the FWI

*The Fire Weather Index system provides an indication of fire danger conditions as influenced by four weather parameter, temperature, relative humidity, precipitation and wind speed \citep{vanwagner:87}.*

Both FWI and GWIS do not apply any overwintering, the FWI system is run in a continuous loop.

*ECMWF implementation for the FWI is initialised once starting from idealised conditions following \cite{wotton:09} values. It also does not implement any overwintering meaning that the moisture codes are not reset to zero during cold winter months.*

L80: Abatzoglou et al. (2018) showed that the FWI is correlated with burned area across some, rather than most, non-arid regions. Please list here which regions those were.

We have been more informative in this citation and added more details as suggested. This paragraph now reads:

*\cite{abatzoglou:18} showed that FWI exhibits strong correlative relationships to burned area across some non-arid eco-regions globally albeit only weaker relationships in climatically drier regions (shrubland) with the larger correlation found in the boreal and evergreen temperate forests of western North America. Also \citet{bowman:17} highlighted how high FWI values are often associated with the most extreme fire activities recorded using Fire Radiative Power observations.*

L96: by 'time strips', do you mean 'time steps'?

Time strips are the longitudinal location of the time steps as per the figure attached. A 24 hours forecast is used to construct a snapshot of atmospheric conditions at local noon. We have explained this more clearly in the text

[Figure]

L105: suggest replacing 'substitutes' with 'replaces'.

Done

L111: as far as reanalysis being a good proxy for meteorological observations, please mention the extent to which is the case for the ECMWF products, drawing on Vitolo et al. (2019) for ERA-I, and how this might have changes for ERA5.

This is a good point and era-5 based FWI reanalysis has been fully validated in a paper that is under review in the Scientific Data journal. Considering the timing that paper will be published ahead of this and probably could be fully referenced. For now we have added a reference as submitted as follows

*A full validation of the FWI database derived from ERA5 can be found in \cite{vitolo2020}*

L153: suggest changing 'if there is no ignition' to 'if there is no ignition and/or aggressive fire suppression'.

Changed

L180: throughout, the BOAS region is missing from the analysis. Was there a reason for this, or should it be added?

We refer to Boreal regions meaning both BOAS and BONA but this might not be clear to the reader in fact. We have now specified this in the text. We did not have SYNOP stations in the BOAS region. Therefore BOAS validation is missing in Figures 2 and 4. However we had FRP observation and included BOAS in the Figure 6.

L182: As far as performance degradation goes, in Figure 2, the differences between ERA5 and the forecasts can be seen, and the differences between regions can be seen, but the panels in Figure 2 are too small to be able to see any performance degradation with increasing lead time.

This is a fair point, however the degradation with lead time is much smaller than the bias bar. A comment on this has been added to the text

*"As expected there is a performance degradation going towards longer lead times however the increase is within the distribution and mean biases are limited to few units even at day 10."*

L194: In Figure 3, is there a reason for the discontinuity between day-6 and day-7 lead times? Similarly for the more apparent discontinuities in Figure 4 for BONA, SEAS. And for TENA, what is the possible reason why the CRPS is lower at lead 7 than lead 6

This is a very good point and at lead time 6 we have a change of time resolution in the forecast when the 3 hourly forecast becomes 6 hourly. This has to be explained and sorry if we missed this in the first version of the paper

*"The discontinuity visible at day 6 is finally an artefact due to the change in temporal resolution in the ecmwf forecast. Up to day 6 forecasts are stored 3 hourly and only 6 hourly after this."*

L199: Change 'Boreas' to 'Boreal'.
Done

L199-L204: It is hard to understand how FWI calibration (in the sense of interpreting it as a fire danger metric) influences forecast performance for different regions. Instead, the differences are more likely due to performance in forecasting the temperature, relative humidity, wind speed and precipitation input variables. Here, please describe possible drivers of these differences in terms of known differences forecast skill for the four inputs, which I trust are evaluated routinely at the ECMWF.

Yes the sentence was very badly expressed and CRPS is not dependent on the quality of the FWi as fire model but on the quality of the inputs. This section has been widely rewritten and extended. It now reads as follow:

*There are some regional differences in the skill provided by the ensemble forecast. Regions covered by Boreal forests (e.g. BOAS, BONA, CEAS) have the largest predictability with the maximum gaps between the forecast and the climate CRPS scores (Figure~\ref{fig:crp_region}). Savannah regions (NHAF, AUST,SHAF) tend to have a shorter window of predictability with the forecast CRPS curve approaching at a shorter lead time than the CLIM ones. The regional differences in the prediction of the forecast FWI when compared to ERA5 derived databases are related to the skills of the forecast which then project in the accuracy in the FWI simulation. While temperature predictions skills are globally mostly uniform, a complex picture emerges for the forecast skills of precipitation in most of the global models used for numerical weather prediction including ECMWF model. Prediction of precipitation in the mid latitudes is notoriously more accurate than in tropics due to its connection with frontal systems driven by large scale dynamics \citep{simmons:02} when compared convective precipitation which is the main source of rainfall in the tropics. Although the gap has been filled through the years forecast predictions in the Southern extra-tropical region is less accurate than the equivalent in the Northern hemisphere due to the availability of a better observing system to constraint the forecast initial conditions \citep{p19277}. These considerations could largely explain the better performances of the FWI predictions in the northern hemisphere for the year taken in consideration. However it has to be noted that forecast skills have strong year to year variations with expected increased skills in the tropic when large scale phenomena such as the Madden-Julian Oscillation (MJO) and /or the El N\~ino Southern Oscillation (ENSO) take place. Under these phenomena the predictability of the tropics and of extratropical regions can substantially improve through teleconnections \citep{vitart:14}.*

L217: BONA is by definition is not a temperate region, please correct.
Yes sorry, changed

L220: change 'false-alarm' to 'false-alarms'?
Changed

L222: Reference required for the statement of weather playing a less relevant role compared to fuel availability in this region.
A reference to krawchuk PLOSONE 2009 has been added

L225: the de Groot et al. (2007) study is only relevant to the EQAS region, not SEAS, and despite recommending other FWI subcomponents, does not appear to say that the FWI is not a good indicator in the area – please correct. Furthermore, while fires in this area are related to land clearing, the Abatzoglou et al. (2018) study shows that the FWI is highly correlated with burned area over EQAS at longer time scales, and as strongly as anywhere else in the world. The much more likely reason for the poor performance was that 2017 was a record low fire year over EQAS (see GFED dry matter emissions estimates here: https://www.geo.vu.nl/~gwerf/GFED/GFED4/tables/GFED4.1s_DM.txt), and there was simply too little fire activity to make a meaningful prediction, as it was probably too wet for any serious burning. Please correct this, ideally with a brief mention of whether the FWI was anomalously low relative to the ERA5 FWI climatology.

This is a very valuable comment of which I hadn't thought. Also in the ERA5 paper that we have reviewed and hopefully will be accepted in print before this one, so it can be referenced, there is a figure that exactly shows the point made on the low fire activities in the EQAS area in 2017. So thanks a lot for the suggestion. The reasoning of the low score in EQUAS has been revised and reads as follows:

*One important exception is the very low performance of the fire forecast in Equatorial Asia (EQAS) and South East Asia (SEAS) where the system seems to have a predictability below 0.2 (only 20 \% of fires corresponded to FWI above the 90th percentile). \citet{degroot:07} highlighted how FWI is not the best indicator in the EQAS areas and a fire early warning system should mostly rely on the drought code. There are a number of factors that could contribute to this low usability of the FWI in these areas. Fires in these regions are mainly caused by humans for the purposes of cleaning the land for establishing plantations \citep{field:09,benedetti:16} and weather, which is the only driver of the FWI, is not the main fire trigger. However it has to be noted that 2017 was a very wet year in EQAS and anomalously low FWI were predicted (see Figure 7 in \cite{vitolo:20} ) with a consistent low emissions recorded by GFED. The low level for fire activities in 2017 means that the applicability of the results for this region in 2017 might not extend to other years with stronger activities.*

L239: referring to the GFED emissions tables above, 2017 was not extreme across the globe. Please correct.

Thanks this has been clarified now and reads as follows

*Here we will analyse three cases of fire events that took place in 2017, which proved to be an year with extreme fire episodes across the globe.*